# A New Cold-Adapted and Salt-Tolerant Glutathione Reductase from Antarctic Psychrophilic Bacterium *Psychrobacter* sp. and Its Resistance to Oxidation

**DOI:** 10.3390/ijms21020420

**Published:** 2020-01-09

**Authors:** Yatong Wang, Quanfu Wang, Yanhua Hou

**Affiliations:** 1School of Environment, Harbin Institute of Technology, Harbin 150090, China; wangyatong199311@163.com; 2School of Marine Science and Technology, Harbin Institute of Technology, Weihai 264209, China; marry7718@163.com

**Keywords:** glutathione reductase, cold-adapted, Antarctic, antioxidant defense, homology modeling

## Abstract

A new glutathione reductase gene (*psgr*) coding for glutathione reductase (GR) from an Antarctic bacterium was cloned and overexpressed into *Escherichia coli* (*E. coli*). A sequence analysis revealed that PsGR is a protein consisting of 451 amino acids, and homology modeling demonstrated that PsGR has fewer hydrogen bonds and salt bridges, which might lead to improved conformational flexibility at low temperatures. PsGR possesses the flavin adenine dinucleotide (FAD) and nicotinamide adenine dinucleotide phosphate (NADPH) binding motifs. Recombinant PsGR (rPsGR) was purified using Ni-NTA affinity chromatography and was found to have a molecular mass of approximately 53.5 kDa. rPsGR was found to be optimally active at 25 °C and a pH of 7.5. It was found to be a cold-adapted enzyme, with approximately 42% of its optimal activity remaining at 0 °C. Moreover, rPsGR was most active in 1.0 M NaCl and 62.5% of its full activity remained in 3.0 M NaCl, demonstrating its high salt tolerance. Furthermore, rPsGR was found to have a higher substrate affinity for NADPH than for GSSG (oxidized glutathione). rPsGR provided protection against peroxide (H_2_O_2_)-induced oxidative stress in recombinant cells, and displayed potential application as an antioxidant protein. The results of the present study provide a sound basis for the study of the structural characteristics and catalytic characterization of cold-adapted GR.

## 1. Introduction

In low temperature, psychrophilic bacteria experience a lot of cold-induced stress. Low temperatures impose strict physical and chemical constraints on membrane fluidity, enzyme kinetics, and macromolecular interactions [1]. As a result, psychrophiles have evolved mechanisms that enable them to successfully combat other stressors associated with cold environments, such as dryness, radiation, an excessive amount of UV radiation, and high salinity [2,3]. Glutathione reductase (GR) (EC 1.8.1.7) belongs to the family of flavoprotein oxidoreductases, provides an NADPH-dependent catalytic reduction of GSSG to the reduced GSH form, and is considered to be a crucial enzyme that maintains the redox status of the cell during oxidative stress [4,5]. GR plays an important role in resisting oxidative stress and maintaining the reducing ability of cells through intracellular GSH/GSSG balance. Furthermore, GR plays a vital role in response to stress, including oxidative stress [6,7], chilling stress [8], salt stress [9], and metal tolerance to ions [4]. Due to their importance, GRs have been cloned and purified from many plant and microbial sources. In order to explain its structure, kinetic mechanisms and properties, a great deal of research has been carried out. Low temperatures, high temperature, drought stress, and salt stress may have an effect on the expression of GR [6,10]. A structural characterization of GR from *E. coli* has been carried out [11]. GR from *Streptococcus pneumoniae* (*S. pneumoniae*) was overexpressed and purified and its crystal structure was determined [4]. Regarding the enzyme’s structure, most GR proteins of prokaryotes and eukaryotes exist as dimers [4,12]. GR from *S. pneumoniae* has a dimeric structure that includes an antiparallel beta-sheet at the dimer’s interface [4]. Two forms of GR have been identified in plants: An isoform that is localized to both chloroplasts and mitochondria (GR1); and a cytosolic isoform (GR2) [13]. Compared with the GRs from plants, we have less of an understanding of GRs from extreme microorganisms.

In recent years, the Antarctic has attracted attention due to its special climate [14,15]. The unique biochemical functions of cold-adapted enzymes play an important role in the adaptation of sea-ice bacteria [16,17]. Based on the importance of GRs in the cellular physiology and biochemistry related to cold adaptation, and to discover and understand their biochemical properties, cold-adaptation and structure deserve further investigation. Therefore, it is necessary to obtain information on the structural characteristics and catalytic properties of GR. *Psychrobacter* sp. strain ANT206 is a psychrotrophic bacterium that has been isolated from Antarctic sea-ice; the thiol-dependent antioxidant protein peroxiredoxin from this strain also exhibited special catalytic properties and has the function of protecting super-coiled DNA from oxidative damage [17]. Here, we isolated a new *psgr* from an Antarctic psychrophilic bacterium. Then, we analyzed *psgr* by its expression in *E. coli*, and the recombinant GR was studied in terms of its molecular and biochemical properties and oxidative resistance.

## 2. Results and Discussion

### 2.1. Identification of the psgr Gene

Sequence analysis of the *psgr* gene revealed an open reading frame (ORF) finder of 1356 bp and *psgr* encoded a protein consisting of 451 amino acids. The GR from *E. coli*, *Colwellia psychrerythraea* (*C. psychrerythraea*), *Sphingomonas* sp. (*S.* sp.), *Pseudomonas aeruginosa* (*P. aeruginosa*) encoded 450, 454 [6], 449 [18], and 451 [19] amino acids, respectively. A multiple sequence alignment of PsGR showed that it is most similar to *Psychrobacter arcticus* GR (90.69%), followed by the *E. coli* GR (75.17%) and the *Streptococcus thermophilus* (*S. thermophilus*) GR (67.18%). In addition, alignment of the amino acid sequence revealed that it has one FAD binding motif as well as one NADPH binding motif (Figure 1). Furthermore, PsGR was found to have the FAD binding motif (GXGXXGX_17_E) and the NADPH binding motif (GXGYIAX_18_RX_5_R) that are typically present in the other GR homologs, such as the GR from *Psychrobacter* sp. YGAH215 and *E. coli*.

### 2.2. Homology Modeling Analysis and Binding Energy Interactions of PsGR

A Ramachandran plot analysis was used to verify the 3D model, and the results indicate that the 3D model of PsGR has excellent eligibility. Hydrogen bonds and salt bridges are frequently found to improve an enzyme’s stability, and there are usually lees of them in cold-adapted enzymes [20]. To identify the cold-adapted structural characteristics of PsGR, a structure model of PsGR and a structure model of its homologue EcGR (*E.coli* GR, 1GET) were compared and found to superimpose well (Figure 2). According to Table 1, PsGR has fewer hydrogen bonds and salt bridges than EcGR, leading to decrease in the stability and thermal stability of this cold-adapted protein [20]. In addition, PsGR had fewer hydrophobic interactions compared to EcGR, which may reduce the rigidity of PsGR, resulting in decreased structural stability [21]. Overall, the sequence of PsGR might increase its conformational flexibility and low temperature catalytic competence, which might contribute to the cold adaptation of rPsGR. The antioxidant capacity of PsGR was evaluated by PsGR’s affinity for GSSG and NADPH as, compared to the interaction of EcGR (Table 2). As can be seen, PsGR exhibits a higher affinity for NADPH and GSSG than the EcGR. A similar result was obtained for GR from *Streptococcus thermophilus* [22]. The value of the binding energy interactions of PsGR via NADPH and GSSG indicted that PsGR has a higher binding capacity for NADPH than for GSSG.

### 2.3. Expression, Purification, and Enzyme Assays of PsGR

rPsGR with an approximately molecular mass of 53.5 kDa was found to expressed in *E. coli* BL21 (Figure 3, Lane 3), *S.* sp. GR and *C. psychrerythraea* GR had a molecular weight of ~50 [18] and 48.7 kDa [6], respectively. After purification by Ni-NTA affinity chromatography, SDS-PAGE showed a main band of rPsGR. Additionally, the purity of rPsGR was approximately 5.6-fold and the recovery rate was 33.3%. The purity and recovery rate of GR from *Enterococcus faecalis* (*E. faecalis*) and *E. coli* were found to be 634-fold and 45% [23] and 58.3-fold and 74% [24]. The specific activity of rPsGR was 168.8 mmol/min/mg, which is higher than that of *E. faecalis* GR (34.43 μmol/min/mg) [23] and *Rhodospirillum rubrum* (*R. rubrum*) GR (102 μmol/min/mg) [25].

### 2.4. Biochemical Characteristics of rPsGR

Table 3 provided a comparison of information on GRs from prokaryotic origin. rPsGR was mostly active at 25 °C (Figure 4a), which is consistent with data from the Antarctic microalgae *Chlamydomonas* sp. [26]. Unexpectedly, GR from the Arctic bacterium *S.* sp. exhibited optimum activity at 60 °C [18]. The optimal temperature of GR from *Phaeodactylum tricornutum* [27] and *Pennisetum glaucum* [12] were 32 and 40 °C, respectively. It is worth noting that the rPsGR activity was 42.2% of its highest activity at 0 °C. However, for GR from *Chlamydomonas* sp. 23.5% of the highest activity remained at 0 °C [26], suggesting that rPsGR is a cold-adapted protein. Regarding thermostability (Figure 4b), the thermal half-life of the rPsGR and *C. psychrerythraea* GR were 30 and 40 min [6] at 50 °C, respectively. The half-life of *E. coli* GR was found to be 2 min at 37 °C [28]. In addition, rPsGR lost 80% of its activity after incubation at 55 °C for 30 min, however, for GR from *Chlamydomonas* sp. approximately 60% of its initial activity remained after incubation at 55 °C for 30 min [26]. rPsGR completely lost activity after incubation at 55 °C for 60 min. However, pure GR from *E. coli* was quite heat stable, denaturing significantly only after incubation for 10 min at 70 °C [29]. *S.* sp. GR was denatured after incubation at 80 °C for 1 h [18]. These results indicate that rPsGR is a cold-adapted enzyme, which is consistent with the results of the homology modeling, and could be used to maintain the GSH/GSSG balance in low temperature environments.

On the basis of the above results, the optimal pH for rPsGR activity was analyzed at 25 °C (Figure 4c). The investigation of the effect of pH showed that rPsGR was active over a broad pH range. rPsGR activity was found to be optimum at 7.5. Similar results have previously reported for *S.* sp. GR [18] and *E. coli* GR [29]. Different results have previously been reported for recombinant GR expressed in *Chromatium vinosum* (*C. vinosum*) GR (pH 7.0) [30]. After incubation for 30 min in buffers with different pH values for the purpose of a pH stability test (Figure 4d), rPsGR activity displayed a downward trend, and remained stable at pH 7.0–8.0. The GR from *E. coli* S33 was stable at pH values ranging from 7.5 to 9.5 [29]. rPsGR maintained 53.9% of its full activity at a pH of 5.0 after incubation for 30 min. *Chlamydomonas* sp. GR only lost 10% of its highest activity at a pH of 5.0 after incubation for 30 min [26]. rPsGR activity decreased when the pH value was higher than 8.0. The activity of GR from *Phaeodactylum tricornutum* was found to decrease significantly above a pH of 8.5 [27]. Moreover, a significant decrease in *E. coli* S33 GR activity was noticed at pH values below 7.5 [29], and the *E. coli* GR showed almost complete inactivation at pH values below 5.5 and over 7.5 [28].

Based on the above described single-factor experimental results, we used the optimal enzyme activity assay system (with the standard method) to study the effects of various factors on enzyme activity. Enzyme activity was affected by the presence of various NaCl concentrations (Figure 4e). The maximal activity of rPsGR was found to occur in 1.0 M NaCl. However, GR from *Setaria cervi* bovine filarial worms were found to exhibit maximum activity between 0 and 0.2 M NaCl [31], which is a lower salt-tolerance than that of rPsGR, indicating that rPsGR is a salt-tolerant enzyme. Taken together, these results suggest that rPsGR could be an important enzyme that is able adapt to the Antarctic sea-ice environment, which has a high salt concentration (2.5 M) [32].

The effect of different additives on rPsGR activity were measured (Table 4). For the purified rPsGR, there was a slight increase in activity with the addition of 0.25 mM MgCl_2_. This increase in activity suggests that Mg^2+^ is important to the activity, similar results were obtained for the GR from *Chlamydomonas* sp. [26]. Furthermore, KCl (0.25 and 1 mM) and CaCl_2_ (0.25 mM) slightly inhibited rPsGR activity. K^+^ had also been found to inhibit *Setaria cervi* GR activity [31]. Additionally, the present of PbCl_2_, CuCl_2_, and HgCl_2_ strongly inhibited the activity of rPsGR. The addition of 0.25 mM CuCl_2_ to *C. vinosum* GR inhibited its activity by 33% [30]. The addition of 0.25 mM ZnCl_2_ to rPsGR and *C. vinosum* GR inhibited their activity by approximately 67% and 10% [30], respectively. The GR activity was only 10.9% of the control’s activity after treatment with 0.25 mM HgCl_2_. Pb^2+^ and Cu^2+^ were found to have significant inhibitory effects on rPsGR. These two metal ions were also found to inhibit the activity of GR from *Chlamydomonas* sp. [26].

### 2.5. Kinetics and Thermodynamics Parameters

The kinetic parameters of rPsGR were determined with GSSG and NADPH as substrates at 25 °C (Figure 5). The *V*_m_ for GSSG was 172.41 mmol/min/mg (Table 5), and *V*_m_ for GR from *R. rubrum* using GSSG as a substrate was 102 U/mg [25]. In general, *K*_m_ can regulate a cold-adapted enzyme’s activity to confer flexibility upon it, and the temperature dependence of the *K*_m_ value optimizes the kinetics of the enzyme to suit its environmental conditions [33,34]. At 25 °C, the *K*_m_ for NADPH was found to be 16.95 μM, which is higher than the *K*_m_ for GR from *C. psychrerythraea* (10.9 μM), mesophilic baker’s yeast (3.9 μM) [6], and *E. coli* K-12 (4.8 μM) [35]. The *K*_m_ for NADPH from *S. pneumoniae* (23.2 μM) [4], *E. coli* GR (25 μM) [24], and *Xanthomonas campestris* (*X. campestris*) (52.6 μM) [5] also have been reported. In addition, the *K*_m_ for GSSG was 68.96 μM, which is lower than GR from *C. vinosum* (7 × 10^3^ μM) [30]. Furthermore, the results showed that GR, such as most psychrophilic enzymes, has high activity at low temperatures, but at the cost of substrate affinity [36]. The *K*_m_ for NADPH (16.95 μM) was lower than the value of GSSG (68.96 μM). This result indicates that the rPsGR had a higher substrate affinity for NADPH than for GSSG, which is consistent with the general trend [4]. Formally, *k*_cat_ is a fundamental kinetic parameter that characterizes an enzymatic reaction and the enzymatic reaction rate is given by the catalytic constant *k*_cat_ [30]. The *k*_cat_ for rPsGR at 25 °C was 94.97 1/s. A *k*_cat_ for *E. faecalis* of 145 1/s [23] and *C. psychrerythraea* of 700 1/s have also been reported [6].

Thermodynamic parameters of the rPsGR were calculated at different temperatures (Table 6). The free energy of activation, ΔG, can be used to analyze the thermostability of enzymes [37]. The higher Δ*G* is, the more stable the enzyme is. The ΔG values of the NADPH substrate increased from 60.07 to 61.80 kJ/mol as the temperature increased from 0 to 25 °C, suggesting that the thermostability of the rPsGR increased as the temperature increased. This trend was also observed in other cold-adapted enzymes [16,38]. Additionally, the enthalpy of activation, Δ*H*, showed a downward trend from 0 to 25 °C. The Δ*H* of GR from Red Spruce was found to be 54 and 43 KJ/mol at 10 and 35 °C, respectively [39], without significant variation. It is known that cold-adapted enzymes have in common the property of a lower Δ*H*, which allows them to increase *k*_cat_ at low temperatures [33]. It has been speculated that rPsGR had a smaller Δ*H* as a cold-adapted xylanase [38]. Notably, the enzyme had a *k*_cat_ value at 0 °C that was approximately 4.98 times lower than that obtained at 25 °C, and *k*_cat_ value of GR from Red Spruce at 10 °C was found to be eight times lower than that at 35 °C [39]. Therefore, the kinetic and thermodynamic results demonstrate that rPsGR is an enzyme adapted to low temperatures.

### 2.6. Disk Diffusion Assay

To determine the oxidation resistance of GR in recombination strains, the sensitivities of BL21/pET-28a(+) and BL21/pET-28a(+)-rPsGR to H_2_O_2_ were performed by a disk diffusion assay (Figure 6). After treatment with 3.0 μL of 30% H_2_O_2_, the clearance zones in the recombinant bacteria (on the plates with a diameter of 2.0 cm) were significantly smaller than those of the controls (on the plates with a diameter of 4.0 cm). These results demonstrate that rPsGR had an antioxidant effect and overcame the oxidative stress induced by H_2_O_2_. Thus, rPsGR might have application as a potential source of antioxidant proteins in the pharmaceutical and food industries. Similarly, GR from the oxygen-tolerant *Streptococcus mutans* was also shown to protect cells from oxidative stress induced by H_2_O_2_ [7].

## 3. Materials and Methods

### 3.1. Bacterial Strains and Plasmids

The strains used in this study are listed in Table 7.

*Psychrobacter* sp. ANT 206 (GenBank accession numbers MK968312), isolated from a sample of Antarctic sea-ice (68° 30’ E, 65° 00’ S), was cultured in 2216E sea water medium (5 g of peptone (Sinopharm, Beijing, China) and 1 g of yeast extract powder (Sinopharm, Beijing, China) per 1 L of solution, pH 7.5) at 12 °C with shaking at 200 rpm and then frozen in glycerol at −80 °C for long term storage. *Psychrobacter* sp. ANT206 was used as a source of chromosomal DNA. All strains were stored in 16% glycerol at −20 °C, BL21/pET-28a(+) and *E. coli* BL21 were cultured in Luria Bertani (LB) medium (10 g of tryptone (Sinopharm, Beijing, China), 5 g of yeast extract powder, and 10 g of NaCl (Sinopharm, Beijing, China) per 1 L of solution, pH 7.0) at 37 °C.

### 3.2. Gene Cloning and Bioinformatics Analysis of PsGR

According to the genomic sequence and the annotation of ANT206 (manuscript in preparation), genomic DNA was used as a template and the following primers were used to amplify the *psgr* coding sequence by PCR: A forward primer with the *Bam*H I site 5′-ATCGGATCCATGACAAAACATTATG-3’ and a reverse primer with the *Xho* I site 5′-TAACTCGAGAGCGCATCGTCACAAA-3′. PCR was performed with Taq DNA polymerase and consisted of initial denaturation at 94 °C for 5 min, and then 30 cycles of denaturation at 94, 59.1, and 72 °C for 1 min, with a final extension at 72 °C for 10 min. The reaction was performed in 20 µL of solution containing 13.8 µL of H_2_O, 2 µL of 10 × PCR buffer (Mg^2+^ plus), 0.125 mM dNTP mixture, 0.5 µM of each primer, 1 µL of template DNA, and 1 U of Taq DNA polymerase. All reagents used in PCR were obtained by TaKaRa Bio (Dalian, China). The complete amino acid sequence and the multiple sequence alignments of PsGR were obtained by ORF and the BioEdit and EScript tools (http://espript.ibcp.fr/ESPript/cgi-bin/ESPript.cgi, last accessed on: 8 January, 2020), respectively.

### 3.3. Homology Modeling and Binding Energy Interactions of PsGR

The 3D structure of the PsGR was obtained using a SWISS-MODEL server. GR (EcGR, PDB ID: 1GET) in *E. coli* capable of encoding 450 amino acids was selected as a template for a 3D model, and the quality of the model was evaluated by a Ramachandran plot analysis. The 3D molecular viewer PyMOL (DeLano Scientific LLC, CA, USA) was used to generate homology model images. Meanwhile, ESBRI (http://bioinformatica.isa.cnr.it/ESBRI/introduction.html, last accessed on: 8 January, 2020) was used to investigate salt bridges, and protein electrostatic interactions were predicted by the Protein Interactions Calculator program (http://pic.mbu.iisc.ernet.in/job.html, last accessed on: 8 January, 2020). The 3D structures of the NADPH and GSSG ligands were obtained from the Zinc database (http://zinc.docking.org/, last accessed on: 8 January, 2020) and Open Babel was used to change sdf files to pdb files. To study the interactions between proteins and ligands, AutoDock Vina [41] by the docking method of grid box parameters was used and run several times. Moreover, the parameters that were adopted for PsGR were center x = 20.000; center y = 32.129, center z = −23.647, size x = 86; size y = 76; and size z = 68.

### 3.4. Expression and Purification of PsGR

The PCR product and the pET-28a(+) vector were digested using the restriction enzyme *Bam*H I (TaKaRa Bio, Dalian, China) and *Xho* I (TaKaRa Bio, Dalian, China) to insert the *psgr* gene, as well as to create a nick in the expression vector, T4 DNA ligase (TaKaRa Bio, Dalian, China) was used to connect the *psgr* gene and pET-28a(+) for 12 h at 16 °C. Then, the ligation system was added to BL21 competent cells (Sangong Biotech, Shanghai, China) and blown evenly. After that, ice bath for 30 min, heat at 42 °C for 90 s, and then placed in an ice bath for 2 to 3 min. Then, it was transferred to 1 mL of LB medium and incubated at 37 °C for 1.5 h. The bacterial solution (200 μL) was applied to a kanamycin (100 µg/mL)-resistant LB solid plate and cultured overnight at 37 °C. Then, the recombinant bacterium was cultured in LB medium at 37 °C until the optical density at 600 nm (OD600) was 0.4–0.6. Expression was induced by addition of 0.5 mM isopropyl β-D-thiogalactoside (IPTG, Solarbio, Beijing, China) and the induction was continued for an additional 24 h at 25 °C. After the cells were induced, they were centrifuged to obtain inclusion bodies. Then, the inclusion bodies were washed twice with 20 mM potassium phosphate buffer (pH 8.0), containing 1 mM dithiothreitol, 1mM EDTA, and 10 μM FAD, and then treated with 8 M urea for 1 h and centrifuged (12,000× *g*) for 15 min, and the supernatant containing crude extract of recombinant PsGR (rPsGR) was recovered. The resulting supernatant was loaded into a Ni-NTA affinity chromatography (GE Healthcare, Uppsala, Sweden) column equilibrated with a buffer (20 mM Tris-HCl, 100 mM NaCl, 10 mM imidazole, pH 8.0), and the proteins were eluted with an elution buffer (20 mM Tris-HCl, 100 mM NaCl, 100 mM imidazole, pH 8.0). Then, the proteins were separated by SDS-PAGE (12.5% polyacrylamide), and the purity and the molecular weight of the rPsGR were estimated.

### 3.5. GR Activity Assays

GR activity was measured by monitoring the oxidation of NADPH in the reaction mixture (200 μL) for 2 min at 340 nm and 25 °C. The reaction mixture contained 100 mM potassium phosphate (Sinopharm, Beijing, China) buffer (pH 7.5), 0.1 mM NADPH (Sigma, Shanghai, China), 1.2 mM GSSG (Sigma, Shanghai, China), and 12.8 µg purified rPsGR. One unit of enzyme activity was defined as the amount of GR that oxidized 1 μmol NADPH per min. The remaining activity was detected using an ultraviolet spectrophotometer (Unico, Okayama, Japan) and a quartz cuvette (Starna, Essex, England) with a path length of 0.1 cm.

### 3.6. Biochemical Characteristics of rPsGR

The purified rPsGR (10.6 µg) was used in a reaction system and the enzymatic properties were investigated. rPsGR activity was determined at different temperatures (0–45 °C) and the optimal temperature of rPsGR was obtained using a standard method described above. The temperature stability of rPsGR was analyzed by pre-incubating the recombinant protein for 60 min at 25 to 55 °C, and the residual activity was detected by the standard method. In order to investigate the optimal pH of rPsGR, two buffers were used for enzyme activity by running the standard assay at 25 °C, and they were the NaAc/HAc buffer (pH 4.5–6.0) and Na_2_HPO_4_/NaH_2_PO_4_ (pH 6.0–8.5) buffer. As for its pH stability, rPsGR was incubated at 25 °C for 30 min in buffers with various pHs, and the remaining activity was tested. To assay its salt tolerance, rPsGR was incubated in 0–3.0 M NaCl at 25 °C for 30 min, and the residual activity was determined. Using the standard method, the effect of adding various reagents of 0.25 and 1 mM (Sinopharm, Beijing, China) was examined. First, KCl, CoCl_2_, MgCl_2_, CaCl_2_, ZnCl_2_, FeCl_2_, CuCl_2_, HgCl_2_, CrCl_2_, CdCl_2_, Pb(NO_3_)_2_, BaCl_2_, and EDTA were mixed with purified rPsGR at 25 °C for 30 min, respectively, and then 100 mM potassium phosphate buffer, 0.1 mM NADPH, and 1.2 mM GSSG were added. The above experiment was carried out three times. Relative activity was expressed as a percentage of the control enzyme’s activity. Furthermore, all temperatures were controlled using a temperature control system JASCO PTC-343 (Jasco International Co., Ltd., Tokyo, Japan). All chemicals were used without further purification.

### 3.7. Kinetics and Thermodynamics Parameters of rPsGR

The kinetic parameters of the purified rPsGR (12.5 μg) were determined from the secondary plots of intercepts versus reciprocal concentrations of GSSG (0.025, 0.05, 0.1, 0.2, 1.0, and 1.2 mM) and NADPH (0.006, 0.013, 0.025, 0.05, and 0.1 mM) at 0–25 °C. Different *k*_cat_ values were obtained at 0–25 °C, and then based on previously described methods [33], the thermodynamic activation parameters (Δ*H*, Δ*S*, and Δ*G*) for rPsGR using NADPH as a substrate were determined based on the *k*_cat_ values. The experiment was carried out three times.

### 3.8. Disk Diffusion Assay

In order to compare the survival rates of *E. coli* BL21 with pET-28a(+) (BL21/pET-28a(+)) and *E. coli* BL21 with pET-28a(+) containing GR (BL21/pET-28a(+)-rPsGR), a disk diffusion analysis was conducted. The IPTG and low temperature (25 °C) induced bacterial cultures were spread on top of LB agar plates. After placing two disks (with a diameter of 3 mm) of Whatman filter paper (Shanghai Jinpan Biotechnology Company Limited, Shanghai, China) on the plate, 1.5 and 3 μL 30% of H_2_O_2_ were added onto the disks. Furthermore, after the plates were treated for 12 h at 37 °C, the cleared zones that reflected the oxidation resistance were measured. The experiment was carried out three times.

## 4. Conclusions

The present study described the cloning of the *psgr* gene of *Psychrobacter* sp. ANT206, overexpression of the protein in *E. coli*, and characterization of the rPsGR. The analysis of the homology modeling showed that PsGR exhibited a decreased number of hydrogen bonds and salt bridges. Therefore, rPsGR showed maximum activity at 25 °C, which proves it to be a cold-adapted enzyme. It was shown to have a high salt-tolerance (1.0 M NaCl). Thus, rPsGR is resistant to low temperatures and high concentrations of NaCl. In addition, the GR in recombinant strains might provide protection to these strains against H_2_O_2_-oxidative stress. Taken together, our results indicate that rPsGR is a novel cold-adapted GR and further reveals its structural characteristics and catalytic properties.

## Figures and Tables

**Figure 1 ijms-21-00420-f001:**
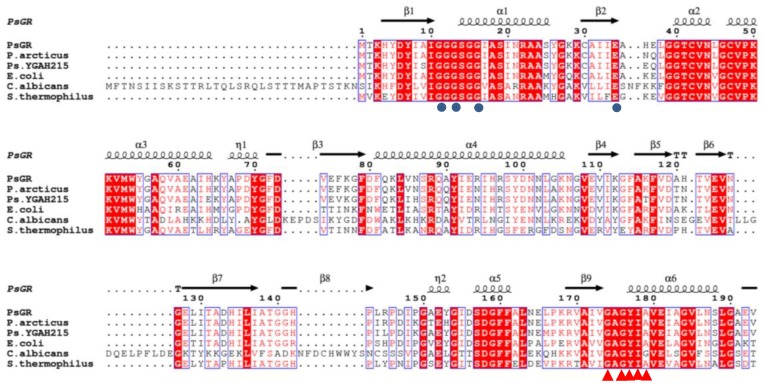
Alignment of the amino acid sequence of *Psychrobacter* sp. ANT206 GR with other glutathione reductases (GRs) from various organisms. PsGR: *Psychrobacter* sp. ANT206 (MN331656) GR; *P. arcticus*: *Psychrobacter arcticus* 273-4 (CP000082); *Ps.* YGAH215: *Psychrobacter* sp. YGAH215 (WP144296620); *E. coli*: *Escherichia coli* (1GET); *C. albicans*: *Candida albicans* (AJ717665); *S. thermophilus*: *Streptococcus thermophilus* (L27672). Flavin adenine dinucleotide (FAD) binding motif represented by blue circles below the *S. thermophilus* sequence; nicotinamide adenine dinucleotide phosphate (NADPH) binding motif represented by red triangles below the *S. thermophilus* sequence.

**Figure 2 ijms-21-00420-f002:**
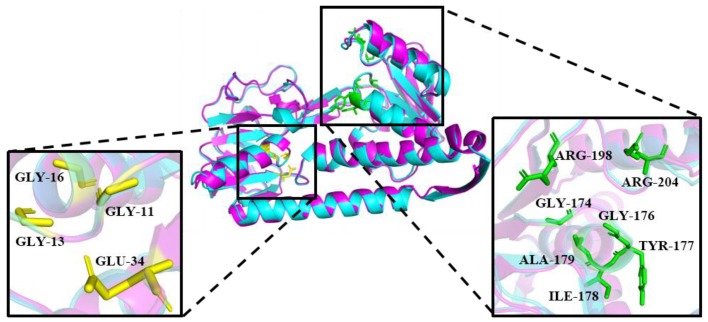
Three-dimensional (3D) structure model of PsGR (purple) superimposition with EcGR (blue, PDB ID: 1GET). The FAD and NADPH binding motifs are indicated as stick models colored in yellow and green, respectively.

**Figure 3 ijms-21-00420-f003:**
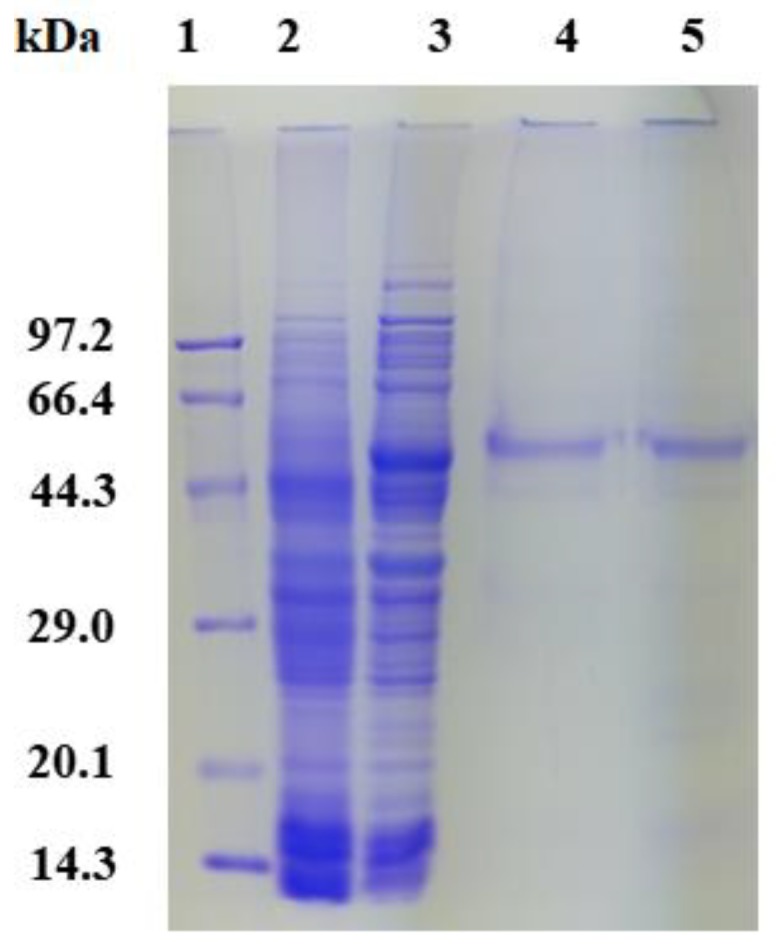
SDS-PAGE analysis of expression and purification of the recombinant PsGR (rPsGR). Lane 1, protein molecular weight marker; Lane 2, crude extract from the BL21/pET-28a(+); Lane 3, crude extract from the BL21/pET-28a(+)-rPsGR; Lanes 4 and 5 purified rPsGR after Ni-NTA chromatography.

**Figure 4 ijms-21-00420-f004:**
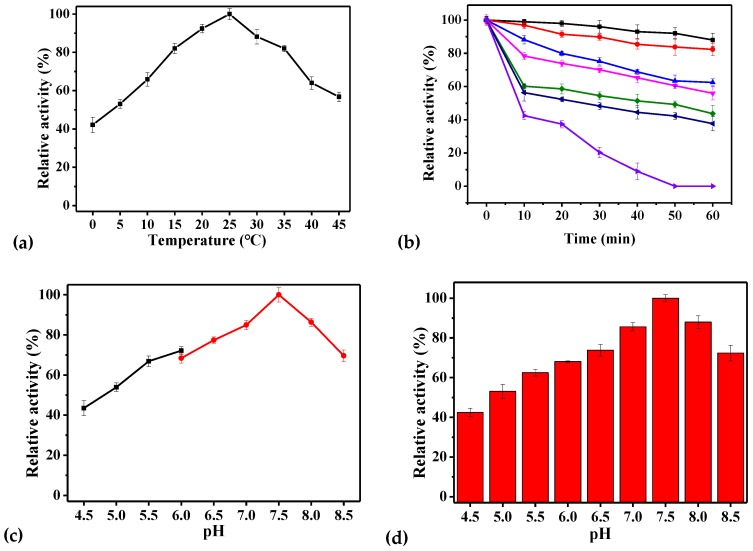
Biochemical characteristics of rPsGR. (**a**) The optimal temperature was determined by measuring the activity at temperatures from 0 to 45 °C. (**b**) Effect of temperatures on the stability of the purified rPsGR. The enzyme was incubated at 25 (■), 30 (●), 35 (▲), 40 (▼), 45 (◆), 50 (◄) and 55 °C (►) for 60 min. Its activity relative to time zero was set as 100%. (**c**) The optimal pH was determined by measuring the activity at pH from 4.5 to 8.5. pH buffer contained NaAc/HAc(■) and Na2HPO4/NaH2PO4 (●). (**d**) Effect of pH on the stability of the purified rPsGR. The enzyme was incubated at 30 °C for 30 min. Its maximal activity was taken as 100%. (**e**) The enzyme was incubated by different concentrations of NaCl for 30 min. Its activity with 0 M NaCl was set as 100%.

**Figure 5 ijms-21-00420-f005:**
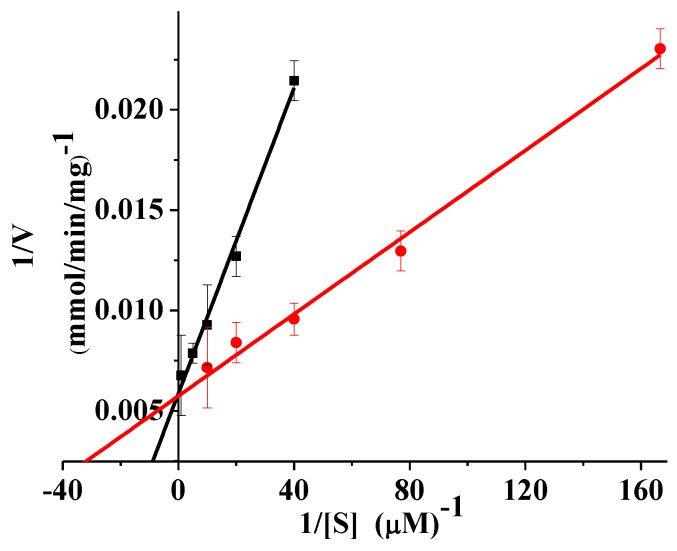
Lineweaver-Burk double reciprocal plots of rPsGR with respect to GSSG (■) and NADPH (●). Activity was measured as standard described. Every experiment was done in triplicates.

**Figure 6 ijms-21-00420-f006:**
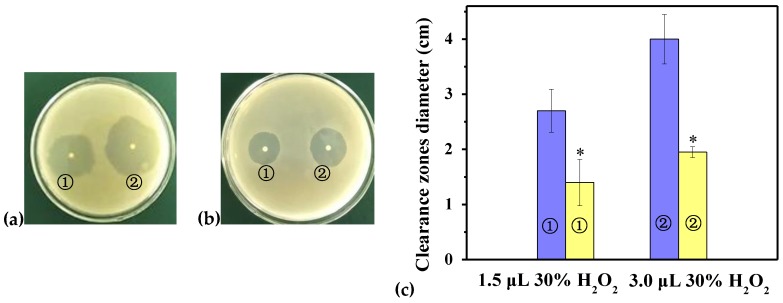
Sensitivity of BL21/pET-28a(+) and BL21/pET-28a(+)-rPsGR to killing by H_2_O_2_. (**a**) The clearance zone diameters (cm) were measured in the plates with BL21/pET-28a(+) after oxidative stress with 1.5 (①) and 3.0 μL 30% H_2_O_2_ (②). (**b**) The clearance zone diameters (cm) were measured in the plates with BL21/pET-28a(+)-rPsGR after oxidative stress with 1.5 (①) and 3.0 μL 30% H_2_O_2_ (②). (**c**) The clearance zone diameters of BL21/pET-28a(+) (blue) and BL21/pET-28a(+)-rPsGR (yellow) after oxidative stress with 1.5 (①) and 3.0 μL 30% H_2_O_2_ (②) were shown in a bar graph form. Data are presented as mean (*n* = 3) ± SD. *, *p* < 0.05, representing a significant difference from the control.

**Table 1 ijms-21-00420-t001:** Comparison of structural adaptation features between PsGR and EcGR.

	PsGR	EcGR	Expected Effect on PsGR
Electrostatic interactions	782	836	Protein stability
Salt bridges	23	32	
Hydrogen bonds	697	741	
Aromatic interactions	32	40	
Cation-Pi interactions	30	23	
Hydrophobic interactions	750	792	Thermolability
G (Gly)	10.4%	9.4%	Flexibility
P (Pro)	3.3%	4.5%	
R (Arg)	3.3%	3.8%	
Arg/(Arg+Lys)	0.34	0.42	
Gly/Pro	3.13	2.09	

**Table 2 ijms-21-00420-t002:** Binding energy interactions of PsGR and EcGR with the NADPH and GSSG as ligands.

Pose Mode	Docking Scores Based on kcal/mol Via the GR-NADPH Interactions		Docking Scores Based on kcal/mol Via the GR-GSSG Interactions	
	PsGR	EcGR	PsGR	EcGR
**1**	−6.9	−5.2	−5.0	−4.8
**2**	−6.6	−5.0	−5.0	−4.7
**3**	−6.5	−5.0	−5.0	−4.5
**4**	−6.3	−4.9	−4.9	−4.5
**5**	−6.1	−4.9	−4.8	−4.3
**6**	−6.0	−4.9	−4.7	−4.3
**7**	−5.8	−4.8	−4.7	−4.3
**8**	−5.8	−4.8	−4.7	−4.2
**9**	−5.6	−4.8	−4.7	−4.2

**Table 3 ijms-21-00420-t003:** Comparison information on GRs from prokaryotic origin.

Source	MW (kDa)	Optimal Temperature (°C)	Optimal pH	Half-Life of Activity (min)	Activator	Km(μM)		*k_cat_* (1/s)	Ref
						GSSG	NADPH		
*E. coli* SG5	49	—	—	—	—	70	25	—	[24]
*E. faecalis*	49	—	—	—	—	80	9	145	[23]
*E. coli* S33	55	—	7.5	—	—	66	16	—	[29]
*X. campestris*	50	—	—	—	—	—	52.6	37.5	[5]
*S.* sp.	50	60	7.5	60 (70 °C)	—	178	—	160	[18]
*C. psychrerythraea*	48.7	—	—	40 (50 °C)	—	—	10.9	700	[6]
*R. rubrum*	54.4	—	—	—	—	90	—		[25]
*S. pneumoniae*	100	—	—	—	—	231.2	23.2	—	[4]
*C. vinosum*	52	—	7.0	—	Na^+^, NH_4_^+^	7000	—	—	[30]
*P*. sp. ANT206	53.5	25	7.5	30 (50 °C)	Na^+^, Mg^2+^	68.96	16.95	95.78	This study

**Table 4 ijms-21-00420-t004:** Effects of different reagents on the rPsGR activity.

Reagent	Concentration	Relative Activity (%)	Reagent	Concentration	Relative Activity (%)
None	—	100 ± 0.0	—	—	—
KCl	0.25 mM	90.3 ± 2.5	KCl	1 mM	81.0 ± 2.0
CoCl_2_	0.25 mM	27.3 ± 3.0	CoCl_2_	1 mM	0.0 ± 0.0
MgCl_2_	0.25 mM	109.1 ± 2.6	MgCl_2_	1 mM	70.3 ± 2.2
CaCl_2_	0.25 mM	90.9 ± 3.1	CaCl_2_	1 mM	78.1 ± 2.5
ZnCl_2_	0.25 mM	33.6 ± 2.5	ZnCl_2_	1 mM	13.4 ± 2.0
FeCl_2_	0.25 mM	44.5 ± 2.2	FeCl_2_	1 mM	5.4 ± 2.7
CuCl_2_	0.25 mM	20.3 ± 2.6	CuCl_2_	1 mM	0.0 ± 0.0
HgCl_2_	0.25 mM	10.9 ± 2.2	HgCl_2_	1 mM	0.0 ± 0.0
CrCl_2_	0.25 mM	47.3 ± 2.9	CrCl_2_	1 mM	15.8 ± 1.9
CdCl_2_	0.25 mM	19.1 ± 2.5	CdCl_2_	1 mM	2.3 ± 3.0
Pb(NO_3_)_2_	0.25 mM	0.0 ± 0.0	Pb(NO_3_)_2_	1 mM	0.0 ± 0.0
BaCl_2_	0.25 mM	37.3 ± 2.7	BaCl_2_	1 mM	8.9 ± 1.6
EDTA	0.25 mM	28.2 ± 2.7	EDTA	1 mM	10.2 ± 2.2

**Table 5 ijms-21-00420-t005:** Kinetic constants of the rPsGR.

Substrate	*V*_m_ (mmol/min/mg)	*K*_m_ (μM)	*K*_cat_ (1/s)
GSSG	172.41	68.96	95.78
NADPH	169.49	16.95	94.16

**Table 6 ijms-21-00420-t006:** Thermodynamic constants of the rPsGR.

Temperature (°C)	△*H* (KJ/mol)	△*S* (J/mol K)	△*G* (KJ/mol)	*K*_cat_ (1/s)
0	38.48	−79.05	60.07	18.90
5	38.44	−78.07	60.15	29.87
10	38.39	−77.44	60.32	44.80
15	38.35	−78.21	60.89	56.12
20	38.31	−79.78	61.70	63.13
25	38.27	−78.93	61.80	94.16

**Table 7 ijms-21-00420-t007:** Strains used in this study.

Strains	Description	Reference
*Psychrobacter* sp. ANT 206	Psychrobacter species	[40]
*E. coli* BL21	*Escherichia coli* BL21	[40]
BL21/pET-28a(+)	*E. coli* BL21 with plasmid pET-28a(+)	[16]
BL21/pET-28a(+)-rPsGR	*E. coli* BL21 with pET-28a(+) containing GR	[16]

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
