# Peer review of "A New Cold-Adapted and Salt-Tolerant Glutathione Reductase from Antarctic Psychrophilic Bacterium Psychrobacter sp. and Its Resistance to Oxidation"

_ijms, 2020, doi:10.3390/ijms21020420_

Round 1
Reviewer 1 Report
General comment
The manuscript ijms-673154 entitled "A New Cold-Adapted and Salt-Tolerant Glutathione Reductase from Antarctic Psychrophilic Bacterium Psychrobacter sp. and its Resistance to Oxidation " by Yatong Wang et al. presents the characterization of a new Glutathione Reductase from a psychrophilic bacterium Psychrobacter sp.. The manuscript presents its purification, its structural model and analyzes its kinetic parameters.
The Glutathione Reductases (GR) play an important role in reaction against diverse oxidative stresses and maintain of the redox balance in all living cells (prokaryotic and eukaryotic ones). The function of the newly characterized enzyme is confirmed by the Disk Diffusion Assay using H2O2 as oxidant. The indisputable interest of the work is the aim of analyzing the structural and kinetic specificities of this enzyme that could be related to its cold-adaptation since its host, Psychrobacter is an Antarctic bacterium. The authors therefore analyze the modelized structure of the enzyme and compare it to the resolved one from Escherichia coli, a mesophilic bacterium. The authors conclude on a correlation between cold-adaptation and the low number of electrostatic interactions, salt bridges and Hydrogen bonds. In a second part of the work the authors analyze several biochemical properties and kinetic parameters of the enzyme: optimal temperature, optimal pH, salt resistance, Vm, Km, Kcat. The authors conclude on a correlation between cold-adaptation and these parameters.
The manuscript describes well done experiments but I have some concerns, however, about the analysis and therefore about the conclusion the authors are drawing. My major concern is about the absence of adequate data taken from the literature to be compared to the presented results with the aim to sustain the interpretation the authors make. And when compared to adequate literature the analyses lack arguments, clarity and rigor. I recommend therefore a major revision of the manuscript.
Specific comments:
-L32-51: In their introduction, the authors exclusively cite works on eukaryotic GRs (refs 1-13). There are, however, numerous works on prokaryotic GRs. Because the presented work relates to a prokaryotic GR, comparison with prokaryotic rather eukaryotic GRs would be more adapted.
-L67-75: As suggested for the Introduction above and as I will also suggest for each result presented in the manuscript, I suggest to compare the newly characterized GR to prokaryotic counterparts. For example, the sequence length of PsGr (451aa) should be compared to the well known E. coli sequence (450 aa) rather than to the one from a plant (Pennisetum glaudum). As exposed by the authors themselves the PsGR shows the highest homology with bacterial GRs rather than eukaryotic one.
The analysis of the sequence motifs is at least awkward. The presence of the conserved FAD and NADPH binding motifs is not specific from psychrophilic organisms but characterize the GR itself. They therefore are conserved in Eukaryotes and Prokaryotes, cold-adapted or not. This is clearly illustrated by the sequence alignment shown in Fig. 1. For didactic purposes I furthermore suggest to present the sequences, in the multiple alignment, by grouping together the prokaryotic sequences.
-L97-99: I do not understand what the authors mean in this sentence. What is consistent with the study of GR from Streptococcus thermophilus? The docking scores via the GR-NADPH and GR-GSSG interactions (see ref 22) are not identical for the common reference system: EcGR.
-L114-118: Again, I suggest to compare the results of expression, purification and enzymology to prokaryotic recombinant GRs in E. coli. The results are more comparable to the one obtained with eukaryotic GRs expressed in E. coli due to the phylogenetic proximity of the host and the parent organisms. There are numerous references of such studies.
-L124-130: I suggest to add to the comparison 1) several other systems between psychrophilic and non psychrophilic GR to avoid bias and also 2) prokaryotic systems.
-L130-133: With a single comparison, with Chlamydomonas, the authors conclude on the particular thermal instability of the rPsGR. But how can they generalize with a single comparison? I furthermore wonder if the used phosphate buffer is adequate for a study in temperature since it is not recognized for its pH stability at high temperature.
-L134-142: The discussion about the pH stability would benefit from a global thorough analysis of the literature and a reference to prokaryotic systems with a discussion on the environmental conditions encountered by the parent organism.
-L145-150: The discussion about the salt effect would benefit from a discussion on the environmental conditions encountered by the GR parent organism. The sea contains in fact 600 mM salt.
-L181-182: I don’t understand what the authors mean here. Colwellia, a psychrophilic organism, shows a lower affinity towards NADPH compared to Ps. And to the other hand (see Scrutton et al. 1987) the Km of EcGR for NADPH is comparable to the one of Ps: 16-23 microM(see Scrutton et al. 1987 or Acta Cryst. (2019). F75, 54–61).
-L185-186: the higher affinity of GRs for NADPH compared to GSSG is a general trend see (Acta Cryst. (2019). F75, 54–61).
There is only one Vm defined per enzyme. 172.41 and 169.49 are in fact representing a single value with errors. This is the same for Kcat with 95.78/94.16 representing a single value with errors.
The authors claim that rPsGR shows particularly high Kcat due to its adaptation to low temperature but where are the compared values? The GR from yeast shows a Kcat of 148, the one from A. faecalis shows a Kcat of 145 (Patel et al. 1998) similar to the one form E. coli.
-L196-206: to be convincing the thermodynamic parameters from psychrophilic and non psychrophilic systems have to be compared here. In Table 5 the DeltaH and deltaG values are not significantly varying and this has also been already seen in Hausladen and Alscher (Plant Physiol 1994). And the high increase of Kcat with temperature is common to cold-adapted and non-adapted organisms (see Hausladen and Alscher, Plant Physiol 1994).
-L239-241: The authors have to explain in Material and Methods how the psgr gene has been identified in the genome and they can not simply refer to a manuscript in preparation.
-L275-282: there is no addition of FAD neither in the growth medium, not in the different purification steps. This is astonishing since FAD is usually add to complement the cofactor lost during maturation and/or purification (see Scrutton et al. 1987 or Acta Cryst. (2019). F75, 54–61).
Minor comments:
-L 54: correct “sea-ice bacterium” for “sea-ice bacteria”
-L70: correct “thermophiles” for “thermophilus”
-L94: I suggest to write “the sequence of PsGR” rather than “PsGR”
-L98: I suggest to rewrite “the PsGR exhibit higher affinity to NADPH and GSSG than the EcGR interaction” and I suggest to write “which is consistent” rather than “which consistent”
-L100: please add a point
-L124: please change “moat” for “most” and add “is” in “which consistent”
-L130: please rewrite “while C. psychrerythraea GR were approximately was 40 min at 50°C”
-L147: please add a point before “However”
-L173: please finish the sentence
-L175: please avoid the s in “kinetics” and “thermodynamics”
-L179: please add “is” in “which higher”
-L271: please correct “an kanamycin” for “a kanamycin”
-L273: please correct” cltured” for “cultured”
-L299: please correct “rPsGR was cultured” for “rPsGR was incubated”
.
Author Response
The manuscript ijms-673154 entitled "A New Cold-Adapted and Salt-Tolerant Glutathione Reductase from Antarctic Psychrophilic Bacterium Psychrobacter sp. and its Resistance to Oxidation " by Yatong Wang et al. presents the characterization of a new Glutathione Reductase from a psychrophilic bacterium Psychrobacter sp. The manuscript presents its purification, its structural model and analyzes its kinetic parameters.
The Glutathione Reductases (GR) play an important role in reaction against diverse oxidative stresses and maintain of the redox balance in all living cells (prokaryotic and eukaryotic ones). The function of the newly characterized enzyme is confirmed by the Disk Diffusion Assay using H2O2 as oxidant. The indisputable interest of the work is the aim of analyzing the structural and kinetic specificities of this enzyme that could be related to its cold-adaptation since its host, Psychrobacter is an Antarctic bacterium. The authors therefore analyze the modelized structure of the enzyme and compare it to the resolved one from Escherichia coli, a mesophilic bacterium. The authors conclude on a correlation between cold-adaptation and the low number of electrostatic interactions, salt bridges and Hydrogen bonds. In a second part of the work the authors analyze several biochemical properties and kinetic parameters of the enzyme: optimal temperature, optimal pH, salt resistance, Vm, Km, Kcat. The authors conclude on a correlation between cold-adaptation and these parameters.
The manuscript describes well done experiments but I have some concerns, however, about the analysis and therefore about the conclusion the authors are drawing. My major concern is about the absence of adequate data taken from the literature to be compared to the presented results with the aim to sustain the interpretation the authors make. And when compared to adequate literature the analyses lack arguments, clarity and rigor. I recommend therefore a major revision of the manuscript.
Specific comments:
-L32-51: In their introduction, the authors exclusively cite works on eukaryotic GRs (refs 1-13). There are, however, numerous works on prokaryotic GRs. Because the presented work relates to a prokaryotic GR, comparison with prokaryotic rather eukaryotic GRs would be more adapted.
Response: We appreciate very much for the Reviewer’s good comments. We have added 8 additional prokaryotic GR references to the full manuscript. This part has been added, Please see page 1-2 line 37-47.
-L67-75: As suggested for the Introduction above and as I will also suggest for each result presented in the manuscript, I suggest to compare the newly characterized GR to prokaryotic counterparts. For example, the sequence length of PsGr (451aa) should be compared to the well known E. coli sequence (450 aa) rather than to the one from a plant (Pennisetum glaudum). As exposed by the authors themselves the PsGR shows the highest homology with bacterial GRs rather than eukaryotic one.
Response: We highly agree and appreciate very much for the Reviewer’s nice comments. This section has been modified and added, please see page 2 line 65-73.
The analysis of the sequence motifs is at least awkward. The presence of the conserved FAD and NADPH binding motifs is not specific from psychrophilic organisms but characterize the GR itself. They therefore are conserved in Eukaryotes and Prokaryotes, cold-adapted or not. This is clearly illustrated by the sequence alignment shown in Fig. 1. For didactic purposes I furthermore suggest to present the sequences, in the multiple alignment, by grouping together the prokaryotic sequences.
Response: This part has been modified in this new version, please see page 2 line 71-73; page 3 Figure 1.
-L97-99: I do not understand what the authors mean in this sentence. What is consistent with the study of GR from Streptococcus thermophilus? The docking scores via the GR-NADPH and GR-GSSG interactions (see ref 22) are not identical for the common reference system: EcGR.
Response: This part has been revised in this version, please see page 4 line 93-95.
-L114-118: Again, I suggest to compare the results of expression, purification and enzymology to prokaryotic recombinant GRs in E. coli. The results are more comparable to the one obtained with eukaryotic GRs expressed in E. coli due to the phylogenetic proximity of the host and the parent organisms. There are numerous references of such studies.
Response: We highly agree and appreciate very much for the Reviewer’s nice comments. This section has been modified and added, please see page 5 line 104-110.
-L124-130: I suggest to add to the comparison 1) several other systems between psychrophilic and non psychrophilic GR to avoid bias and also 2) prokaryotic systems.
Response: We have added 8 additional prokaryotic GR references to the full manuscript. But some studies on the enzymatic properties are not systematic. We have tried to increase the content in the discussion section. This section has been revised in this version, please see page 5 line 116-119, 123-124 and table 3.
-L130-133: With a single comparison, with Chlamydomonas, the authors conclude on the particular thermal instability of the rPsGR. But how can they generalize with a single comparison? I furthermore wonder if the used phosphate buffer is adequate for a study in temperature since it is not recognized for its pH stability at high temperature.
Response: We highly agree and appreciate very much for the Reviewer’s nice comments. This section has been added in this version, please see page 5 line 126-128. Furthermore, we incubated Na2HPO4/NaH2PO4 (pH 7.5) for 1 h at 25-55℃, and the results showed that the pH values at different temperatures were between 7.49-7.51. Based on this, we think phosphate buffer can be used in this manuscript, and it was used as a buffer in the thermostability of some enzymes (Yifan Wang et al, Enzyme and Microbial Technology, 2019, 131, 109434; Shaohui Yuan et al, Journal of Molecular Catalysis B: Enzymatic, 2104, 109, 17-23.).
-L134-142: The discussion about the pH stability would benefit from a global thorough analysis of the literature and a reference to prokaryotic systems with a discussion on the environmental conditions encountered by the parent organism.
Response: We appreciate very much for the Reviewer’s suggestion. This part has been added in this version, please see page 5 line 131-132, 133-134; page 5- 6 line 138-139.
-L145-150: The discussion about the salt effect would benefit from a discussion on the environmental conditions encountered by the GR parent organism. The sea contains in fact 600 mM salt.
Response: In this study, Psychrobacter sp.ANT206 isolated from Antarctic sea-ice sample (68°30′E, 65°00′S). In fact salinity of Antarctic sea-ice is approximately 15 % (2.5 M) (Bowman J P et al, Applied and Environmental Microbiology, 1997, 63, 3068-3078). Strain ANT206 can grow well at salinity of 9%. Therefore, 0-3.0 M NaCl concentration was investigated in this study, and this section has been added. Please see page 6 line 147; page 10 line 232-233.
-L181-182: I don’t understand what the authors mean here. Colwellia, a psychrophilic organism, shows a lower affinity towards NADPH compared to Ps. And to the other hand (see Scrutton et al. 1987) the Km of EcGR for NADPH is comparable to the one of Ps: 16-23 microM(see Scrutton et al. 1987 or Acta Cryst. (2019). F75, 54–61).
Response: We appreciate very much for the Reviewer’s good comments. We deleted this conclusion and compared with other GRs. Please see page 8 line 179-182.
-L185-186: the higher affinity of GRs for NADPH compared to GSSG is a general trend see (Acta Cryst. (2019). F75, 54–61).
Response: It has been revised in new version, please see page 8 line 185-186.
There is only one Vm defined per enzyme. 172.41 and 169.49 are in fact representing a single value with errors. This is the same for Kcat with 95.78/94.16 representing a single value with errors.
Response: We appreciate very much for the Reviewer’s nice comments. We have deleted the content of comparing different substrate Vm values in the previous version.
The authors claim that rPsGR shows particularly high Kcat due to its adaptation to low temperature but where are the compared values? The GR from yeast shows a Kcat of 148, the one from A. faecalis shows a Kcat of 145 (Patel et al. 1998) similar to the one form E. coli.
Response: We have deleted related conclusion and compared Kcat with other GRs. Please see page 8 line 188-189.
-L196-206: to be convincing the thermodynamic parameters from psychrophilic and non psychrophilic systems have to be compared here. In Table 5 the DeltaH and deltaG values are not significantly varying and this has also been already seen in Hausladen and Alscher (Plant Physiol 1994). And the high increase of Kcat with temperature is common to cold-adapted and non-adapted organisms (see Hausladen and Alscher, Plant Physiol 1994).
Response: This section has been added, please see page 8 line 200-201; 204-205.
-L239-241: The authors have to explain in Material and Methods how the psgr gene has been identified in the genome and they can not simply refer to a manuscript in preparation.
Response: We appreciate very much for the Reviewer’s comment. At present, the whole genetic data and analysis of Psychrobacter sp. ANT206 are underway, and other gene sequences of this strain are in a state of protection and cannot be published for the time being. In this version we provide the accession numbers of psgr gene sequence and 16SrRNA sequence. Please see page 3 line 76 and page 10 line 232.
-L275-282: there is no addition of FAD neither in the growth medium, not in the different purification steps. This is astonishing since FAD is usually add to complement the cofactor lost during maturation and/or purification (see Scrutton et al. 1987 or Acta Cryst. (2019). F75, 54–61).
Response: This part has been added, please see page 10-11 line 278-280.
Minor comments:
-L 54: correct “sea-ice bacterium” for “sea-ice bacteria”
Response: This word has been modified, please see page 2 line 53.
-L70: correct “thermophiles” for “thermophilus”
Response: It has been revised in this version, please see page 2 line 69.
-L94: I suggest to write “the sequence of PsGR” rather than “PsGR”
Response: This sentence has been modified in this version, please see page 4 line 90.
-L98: I suggest to rewrite “the PsGR exhibit higher affinity to NADPH and GSSG than the EcGR interaction” and I suggest to write “which is consistent” rather than “which consistent”
Response: This part has been rectified, please see page 4 line 93-95.
-L100: please add a point
Response: This part has been added in this version, please see page 4 line 96.
-L124: please change “moat” for “most” and add “is” in “which consistent”
Response: It has been rectified in new version, please see page 5 line 117.
-L130: please rewrite “while C. psychrerythraea GR were approximately was 40 min at 50°C”
Response: This part has been modified, please see page 5 line 123-124.
-L147: please add a point before “However”
Response: The point has been added in this version, please see page 6 line 144.
-L173: please finish the sentence
Response: We revised this in new version, Please see page 7 line 171-172.
-L175: please avoid the s in “kinetics” and “thermodynamics”
Response: The part has been revised in this version, please see page 7 line 174.
-L179: please add “is” in “which higher”
Response: This part has been revised, please see page 8 line 180.
-L271: please correct “an kanamycin” for “a kanamycin”
Response: The word has been modified in this version, please see page 10 line 273.
-L273: please correct” cltured” for “cultured”
Response: This word has been changed, please see page 10 line 275.
-L299: please correct “rPsGR was cultured” for “rPsGR was incubated”
Response: We appreciate very much for the Reviewer’s nice recommendation. We have revised it in this version, please see page 11 line 303.
Reviewer 2 Report
Summary:
In this manuscript the authors describe the identification, expression and purification of glutathione reductase from the psychrophilic bacterium Psychrobacter. Using homology modeling the authors compare the biochemical properties of Psychrobacter GR with GRs from non-psychrophilic sources and identify differences that are consistent with cold adaptation of the enzyme. Recombinant GR from Psychrobacter was purified and characterized, again identifying enzymatic parameters (temperature, salinity, kinetics) that are consistent with a cold-adapted enzyme. Lastly the authors show that the bacteria expressing this enzyme are less susceptible to oxidative stress. Overall, the work does a good job examining this new GR family member and uncovering its cold adaptation features.
Comments:
A careful review of the English language usage will make the biggest improvement for this manuscript. Currently, the reader must infer some of the points being made by the authors because of unusual phrasing in the text.
Minor comments:
How long are the standard reactions being performed in Fig. 4a and 4c? The methods only state temperature. What is the relative activity in Fig. 4b relative to, e.g. was the activity at 25°C set to 100% at each time point, or is the activity relative to time zero? If the latter, why is there no profile for 25°C? In Fig. 4b does the X-axis time indicate reaction time or preincubation time? Line 310-313 – This reviewer suggests including the reciprocal plots that were used to identify the kinetic parameters to provide an estimate of the fit of these curves. Briefly describe how thermodynamic parameters were elucidated from the kinetic parameters. Were the disk diffusion assays also carried out at lower temperatures, e.g. 25°C?Author Response
In this manuscript the authors describe the identification, expression and purification of glutathione reductase from the psychrophilic bacterium Psychrobacter. Using homology modeling the authors compare the biochemical properties of Psychrobacter GR with GRs from non-psychrophilic sources and identify differences that are consistent with cold adaptation of the enzyme. Recombinant GR from Psychrobacter was purified and characterized, again identifying enzymatic parameters (temperature, salinity, kinetics) that are consistent with a cold-adapted enzyme. Lastly the authors show that the bacteria expressing this enzyme are less susceptible to oxidative stress. Overall, the work does a good job examining this new GR family member and uncovering its cold adaptation features.
Comments:
A careful review of the English language usage will make the biggest improvement for this manuscript. Currently, the reader must infer some of the points being made by the authors because of unusual phrasing in the text.
Response: We highly agree and appreciate very much for the Reviewer’s nice comments. We revised the manuscript carefully to avoid the language errors. And we have consulted a professional English language editing services to check the English. And we believe that the language now is acceptable for the review process.
Minor comments:
How long are the standard reactions being performed in Fig. 4a and 4c?
Response: We appreciate very much for the Reviewer’s good comments. The part has been added in this version, please see page 11 line 289.
The methods only state temperature. What is the relative activity in Fig. 4b relative to, e.g. was the activity at 25°C set to 100% at each time point, or is the activity relative to time zero? If the latter, why is there no profile for 25°C?
Response: The relative activity in Fig. 4b is relative to initial activity. Its activity relative to time zero was set as 100%. At different temperatures, as the incubation time increases, rPsGR activity gradually decreases. We describe the effect of temperature on the stability of this enzyme through Figure 4b. Besides, we added profile for 25 °C. Please see page 7 line 158-159 and Figure 4b.
In Fig. 4b does the X-axis time indicate reaction time or preincubation time?
Response: In Fig. 4b, the X-axis time indicated preincubation time.
Line 310-313 – This reviewer suggests including the reciprocal plots that were used to identify the kinetic parameters to provide an estimate of the fit of these curves.
Response: We appreciate very much for the Reviewer’s kind recommendation. This section has been modified and added, please see Figure 5.
Briefly describe how thermodynamic parameters were elucidated from the kinetic parameters.
Response: We appreciate very much for the Reviewer’s good comments and kind recommendation. The part has been added in this version, please see page 11 line 317-318.
Were the disk diffusion assays also carried out at lower temperatures, e.g. 25°C?
Response: We highly agree and appreciate very much for the Reviewer’s nice comments. We actually induced the gene at low temperature 25 °C (please see page 10, line 276-277). Since the control is BL21/pET-28a(+), in order to make it grow better, we set the temperature to 37 °C (please see page 12 line 326-327).
Round 2
Reviewer 1 Report
General comment
The revised manuscript ijms-673154 entitled "A New Cold-Adapted and Salt-Tolerant Glutathione Reductase from Antarctic Psychrophilic Bacterium Psychrobacter sp. and its Resistance to Oxidation " by Yatong Wang et al. presents significant improvement compared to the first submission. The authors followed major suggestions made by the reviewers and first of all compared the presented results with the one obtained with others prokaryotic systems. The present version, however, still lack pertinent discussion of the presented results and the comparison with other data are often restricted to a simple list of values. The present version therefore still lacks significance of content and scientific soundness. I also regret a missing of any editorial care, bordering a reader disrespect. I cannot believe the authors “have consulted a professional English language editing service(s) to check the English ». I recommend therefore a minor revision of the manuscript.
Specific comments:
-in all the manuscript: please correct” Vm of “, “Km of” or “kcat” of” for “Vm for” , “Km for” and “kcat for”.
-p4, L94: please change “than the EcGR interaction” for “than the EcGR”.
-p4, L94: please correct “the similar result” for “similar result”.
-p5, L116: please correct” the comparison information of GRs from” for “the comparison of information on GRs from prokaryotic origin”
-p5, L117: please correct “was most active” for “mostly active”
-p5, L117: please correct” which is consistent with GR from Antartic” for “which is consistent with data from…”
-p5,123: please correct “GR were was 30 min” for “were 30 min…”
-p5, L131: please correct “activity was most actve” for “activity was optimum…” and correct “is similar as GR” for “is similar to GR”.
-p5, L134: please correct “from 7.5-9.5” for “from 7.5 to 9.5”.
-p5, L138: please correct “how ever” for “however”.
-p8, L182: please delete the point in “[23]. also…”.
-p8, L188: please make an editorial/scientific effort in the sentence “And kcat …and…and…” avoiding “and” while using common units allowing the comparison of data.
Author Response
The revised manuscript ijms-673154 entitled "A New Cold-Adapted and Salt-Tolerant Glutathione Reductase from Antarctic Psychrophilic Bacterium Psychrobacter sp. and its Resistance to Oxidation " by Yatong Wang et al. presents significant improvement compared to the first submission. The authors followed major suggestions made by the reviewers and first of all compared the presented results with the one obtained with others prokaryotic systems. The present version, however, still lack pertinent discussion of the presented results and the comparison with other data are often restricted to a simple list of values. The present version therefore still lacks significance of content and scientific soundness. I also regret a missing of any editorial care, bordering a reader disrespect. I cannot believe the authors “have consulted a professional English language editing service(s) to check the English ». I recommend therefore a minor revision of the manuscript.
Response: We highly agree and appreciate very much for the Reviewer’s useful comments. Due to the incomplete study of the GR properties of prokaryotic sources, we added 8 references for the first version, and this version added 4 references as much as possible. We have added the content of the table 3, discussion of the presented results and the comparison with other data, please see page 2 line 68-70; page 4-5 line 114-116; page 5 line130, 140-142, 150-151; page 7 line 180-182; page 8 line 194-197 and Table 3. Besides, we have also added significance of content, please see page 5 line 135-137 and page 9 line 232-233. In addition, we consulted the English language editing service on MDPI to check English (ID: english-15373). And we think the language can now be used in the review process.
Specific comments:
-in all the manuscript: please correct” Vm of “, “Km of” or “kcat” of” for “Vm for” , “Km for” and “kcat for”.
Response: We highly agree and appreciate very much for the Reviewer’s nice comments. This section has been modified, please see page 8 line 189-203.
-p4, L94: please change “than the EcGR interaction” for “than the EcGR”.
Response: This part has been revised in this version, please see page 3 line 99.
-p4, L94: please correct “the similar result” for “similar result”.
Response: This part has been revised in this version, please see page 3 line 99.
-p5, L116: please correct” the comparison information of GRs from” for “the comparison of information on GRs from prokaryotic origin”
Response: This sentence has been modified, please see page 5 line 122.
-p5, L117: please correct “was most active” for “mostly active”
Response: This part has been revised, please see page 5 line 123.
-p5, L117: please correct” which is consistent with GR from Antartic” for “which is consistent with data from…”
Response: This sentence has been modified, please see page 5 line 123.
-p5,123: please correct “GR were was 30 min” for “were 30 min…”
Response: This part has been revised, please see page 5 line 129.
-p5, L131: please correct “activity was most actve” for “activity was optimum…” and correct “is similar as GR” for “is similar to GR”.
Response: This part has been modified, please see page 5 line 140.
-p5, L134: please correct “from 7.5-9.5” for “from 7.5 to 9.5”.
Response: This sentence has been modified, please see page 5 line 145.
-p5, L138: please correct “how ever” for “however”.
Response: The word has been deleted in new version.
-p8, L182: please delete the point in “[23]. also…”.
Response: This sentence has been modified.
-p8, L188: please make an editorial/scientific effort in the sentence “And kcat …and…and…” avoiding “and” while using common units allowing the comparison of data.
Response: This part has been modified, please see page 8 line 202-204.